# Ag^0^–Ginger Nanocomposites Integrated into Natural Hydrogelated Matrices Used as Antimicrobial Delivery Systems Deposited on Cellulose Fabrics

**DOI:** 10.3390/gels10020106

**Published:** 2024-01-27

**Authors:** Florentina Monica Raduly, Valentin Raditoiu, Alina Raditoiu, Maria Grapin, Mariana Constantin, Iuliana Răut, Cristian Andi Nicolae, Adriana Nicoleta Frone

**Affiliations:** 1Laboratory of Functional Dyes and Related Materials, National Research and Development Institute for Chemistry and Petrochemistry—ICECHIM, 202 Splaiul Independentei, 6th District, 060021 Bucharest, Romania; monica.raduly@icechim.ro (F.M.R.); coloranti@icechim.ro (A.R.); maria.grapin@icechim.ro (M.G.); mariana.calin@icechim.ro (M.C.); iulia_rt@yahoo.com (I.R.); ca_nicolae@yahoo.com (C.A.N.); ciucu_adriana@yahoo.com (A.N.F.); 2Faculty of Pharmacy, Titu Maiorescu University, Bd. Gh. Sincai, No.16, 040441 Bucharest, Romania

**Keywords:** nanocomposites, natural gels, silver nanoparticles, hydrogels, natural antimicrobials

## Abstract

In the textile, medical, and food industries, many of the applications have targeted the use of textile fabrics with antimicrobial properties. Obtaining eco-friendly coatings is of wide interest, especially for applications related to wound dressing or to food packaging. In order to obtain coatings with antimicrobial properties through environmentally friendly methods, a series of experiments were carried out on the use of natural polymers loaded with silver nanoparticles. In this study, coatings with antimicrobial properties were obtained by depositing natural composites based on rice flour, carob flour, or alginate on cotton fabrics. These antimicrobial coatings were multicomponent systems, in which the host matrix was generated via hydration of natural polymers. The nanocomposite obtained from the phytosynthesis of silver particles in ginger extract was embedded in hydrogel matrices. The multicomponent gels obtained by embedding silver nanoparticles in natural polymer matrices were deposited on cotton fabric and were studied in relation to nanoparticles and the type of host matrix, and the antimicrobial activity was evaluated. Fabrics coated with such systems provide a hydrophilic surface with antimicrobial properties and can therefore be used in various areas where textiles provide antibacterial protection.

## 1. Introduction

Migration events determined by population growth in recent years and the free movement of people after the fall of several dictatorial systems in Asia and Africa, as well as technological development, have led, among other things, to the appearance of new types of pathogens in certain regions or the increase in the resistance of already known pathogens to classic treatments [1,2,3]. Thus, the search for new types of antimicrobial treatments, such as photodynamic therapy [4,5], disinfection with UV light [6], the combined use of several known antimicrobial agents [7], or finding new compounds with antifungal activity, was considered a priority. From the last category, a series of natural compounds, polyphenolic derivatives, chitosan, tannins, and essential oils were studied [8,9,10,11,12], and new antibiotics [11,13] or nanomaterials, such as graphene oxide, metallic oxides, and metallic particles [8,14,15,16], were synthesized. All these newly discovered antimicrobial agents must be packaged under different formulations or delivery platforms in order to be used in as many fields as possible. The development and diversification of the segment that refers to the materials from which various objects for domestic use or use in specific fields are formed have led to the demand for researchers’ imagination to create and obtain composites that make it possible to apply new compounds with antimicrobial activity on these surfaces. At the same time, the circumstances related to the reduction in pollution and the promotion of the circular economy support the development of new less polluting and cheaper technologies [17,18,19]. To obtain such results, new methods of recovering active products from food waste [20,21] or obtaining new natural binders or biopolymers were sought [22,23,24,25]. In their studies, the researchers’ attention was drawn to flours and starches that contain polysaccharides, gums, protein, lipids with adhesive properties and swelling properties, minerals, polyphenolic derivatives, or flavonoids with antioxidant activity [26]. Cereal flours (wheat, rice, corn) are traditionally used in the processes of obtaining bakery products [27,28,29] and in the field of fabric dyeing [30,31,32]. Their new applications have been introduced in the food industry or in the restoration of paintings as thickening or gelling agents (tapioca, alginate, carob) [29,33], and many of them, such as rice flour and chitosan, are already used in the cosmetic and pharmaceutical fields and carob flour in the biomedical field [26,34].

From the variety of natural polymers presented, in this paper, three types of agents, namely rice flour, carob flour, and alginate, were chosen, which represented the delivery base of Ag^0^–ginger nanocomposites obtained following the process of reducing silver ions through phytosynthesis in the presence of ginger extract. In this sense, following the gelation process of the flours in the presence of water, the Ag^0^–ginger nanocomposites were embedded in hydrogel matrices and resulted in three types of composites that were deposited by printing on cotton fabric. In this way, three types of fabrics (G1, G2, and G3) were obtained, with antimicrobial properties confirmed using silver ion release tests and tests on *Staphylococcus aureus*, *Escherichia coli*, and *Candida albicans* cultures. The realization of these fabrics covered with eco-composites aimed at obtaining, through an environmentally friendly and economic method, cellulosic materials with antimicrobial properties that can be used in the food field for food packaging, in the medical field for dressings, or for specific clothing.

## 2. Results and Discussion

The benefits of natural ginger extract are well-known, and it has been used since ancient times in the field of traditional medicine [35]. Bioactive compounds and polyphenol derivatives extracted most often in alcohol or water and used as reducing systems to obtain silver nanoparticles have been thoroughly studied [36,37,38]. In this work, based on the results from the literature, we chose to synthesize silver nanoparticles according to El Refai’s recipe [38] from aqueous ginger extract and aqueous silver nitrate solution (Figure 1A), the presence of which was confirmed via UV–Vis analysis (Figure 1B). The Ag^0^–ginger nanocomposites were integrated into three types of hydrogel matrices based on rice flour, alginate, and carob flour (Figure 1A(a–c)). The resulting natural hydrogels were deposited by printing on cotton fabric (G1, G2, and G3). In order to study the influence and necessity of the presence of silver nanoparticles in hydrogels rich in bioactive compounds, we compared the G3-0 fabric, which was covered with the same type of composite as G3 but lacked silver nanoparticles, and the G3-1 fabric, which contained twice as much Ag^0^–ginger nanocomposite compared to G3.

### 2.1. Characterization of the Cellulose Fabrics Finished with Natural Hydrogels

#### 2.1.1. FTIR Spectroscopy

The basic structure of cellulose is made up of D-glucose units linked β(1-4) whose infrared spectral signals will be found in the FTIR spectra of the three fabrics finished with eco-composites (Figure 1). The FTIR peaks present at 3330 and 3270 cm^−1^ were attributed to O–H stretching vibrations, while at 2900, 1426, and 1314 cm^−1^, the peaks corresponding to the C–H stretch were identified. The band at 1640 cm^−1^ was attributed to the O–H bond of adsorbed water, while the peaks at 1107, 1052, and 1030 cm^−1^ belong to C–O vibrations and the peak at 1160 cm^−1^ is attributed to the C–O–C stretching mode. The bands shown at 1107 and 895 cm^−1^ are characteristic of the C–O stretching vibration on the polysaccharide skeleton [39]. In the case of the cellulose supports covered with natural hydrogel, in the FTIR analysis of each sample analyzed, the presence of polysaccharides will lead to the broadening of the characteristic O–H stretching bands around 3300 and 2900 cm^−1^, which correspond to C–H stretching vibrations. At the same time, the hydroxyl [40], carboxylic [41], or methylene [26,42] functional groups grafted on the glycoside residues and the auxiliary compounds that are found in the biopolymer component presented characteristic bands of the coating types. Starch is one of the major constituents of rice flour. It has as main components amylose and amylopectin, which give characteristic spectral bands in the range of 1500–800 cm^−1^ (fingerprint region). In the specialized literature, the IR spectrum of rice flour presents bands at 1540 cm^−1^ attributed to the NH bending vibration due to the protein content, at 1640 cm^−1^ due to the HOH bending vibration mode of water [40], and at 1740 cm^−1^ attributed to carbonyl group stretching vibration due to the content of lipids or oils. During the hydration of the starch matrix, the hydrogel spectrum showed an intense band at 1730 cm^−1^, and the bands at 1640 and 1540 cm^−1^ decreased in intensity and broadened, which proved that proteins are involved in the hydration process [27]. The FTIR-ATR spectrum of alginate has a wide band at about 3200 cm^−1^, which was attributed to the overlapping of O–H stretching vibrations due to the strong intra-molecular and intermolecular hydrogen bonds. The two bands at 1600 and 1410 cm^−1^, respectively, correspond to the asymmetrical and symmetrical stretching vibrations of carboxylic ion [41]. The involvement of carboxylic groups in the process of hydration of the alginate matrix is confirmed in the G2 spectrum by the decrease in the intensity of the two characteristic bands. The appearance of an intense band at 1730 cm^−1^, similar to the one observed in the case of G1, can be assigned to the ester groups, as was already reported [43].

The FTIR absorption spectrum of the carob flour has a wide band at 3290 cm^−1^, attributed to the O–H and N–H stretching vibrations of protein groups, while the signals at 2918 and 2874 cm^−1^ correspond to the asymmetrical and symmetrical stretching vibrations of CH_2_. The wide band at 1633 cm^−1^ corresponds to the stretching or bending vibrations of the NH bonds belonging to the protein residues [42]. It is observed that the structure of the carob flour is similar to that of cellulose, so that the hydrated matrix deposited on the textile support (G3) does not bring significant structural changes in the IR spectrum.

In Figure 1, the FTIR absorption spectrum of the Ag^0^–ginger nanocomposite shows at 3270 cm^−1^ the characteristic band of H–O stretching vibration, while the region at 2925–2866 cm^−1^ is attributed to C–H stretching vibrations. The band at 1560 cm^−1^ corresponds to amide groups, at 1380 cm^−1^ is a band which corresponds to the CH stretching vibration, and the band at 1305 cm^−1^ was attributed to the O–H bond. The band at 1066 cm^−1^ is characteristic of the stretching vibrations of the C–O bonds, while the signal at 773 cm^−1^ corresponds to a vibration characteristic of aromatic rings [36].

#### 2.1.2. X-ray Diffraction Analysis of Fabrics Finished with Hydrogel Matrices

The textile materials finished with natural hydrogels show the presence of the characteristic lines of cellulose in Figure 2a, due to its polymorphism, according to data from the literature [44]. The amorphous region in cellulose was signaled by the appearance of a distinctive peak at 2θ = 32°, corresponding to the lattice plane (004), according to the specifications of JCPDS card no. 00-056-1718 for cellulose Iβ. For the fabric (G3) covered with hydrogels containing AgNPs, the XRD diffraction pattern showed the presence of two diffraction maxima at 38° and 44° corresponding to the diffraction planes (111), and (200) confirming the presence of Ag, in a cubic crystalline system (ICDD PDF No. 01-087-0719). Besides these lines, two other diffraction maxima at 26° and 32° corresponding to (110) and (111) diffraction planes were identified and attributed to Ag_2_O in the cubic crystalline system (ICDD PDF. No. 01-078-5867), which was formed after oxidation of the silver nanoparticles.

#### 2.1.3. SEM-EDX Analysis of Fabrics Finished with Hydrogel Matrices

For the study of the natural matrices used as delivery systems, three types of flour characterized by different saccharide structural sequences were chosen (Figure 3). These, following the gelation process, generate networks with their own architecture and specific characteristics of the resulting gels. Thus, starch, the major component in rice, has the property of swelling and can easily retain interstitial water [28], while the carboxylic groups contained on the polysaccharide chain favor cross-linking processes in the presence of divalent ions [30]. Carob flour has a larger ramified polysaccharide sequence, called galactomannan, which, together with the proteins and fibers contained, manifests itself as gum [26]. The SEM images (Figure 4) of the three types of hydrogels loaded with AgNPs, which were deposited on the cellulosic fabrics, were analyzed, and it was observed that the biopolymer through its structure influences the final gelled compositions. In the case of composites G1 and G3, the starch content of rice flour [28] and the galactomannans from carobs [26], respectively, formed homogeneous mixtures, and the silver particles were well-dispersed in the gelled matrices. Therefore, the type of polysaccharides contained and the initial milling treatments to obtain flour directly influence their solubility [26,28], rheology, viscosity, the gelation process, and water absorption [25,26,27,28,29,30]. The architecture of the gels resulting from the hydration of rice and carob polysaccharides favors a good dispersion of the nanocomposite loaded in the natural polymer network.

Moreover, as can be seen in Figure 4, in the case of G2, the hydrogelated matrixinfluences the compatibility between the host alginate and the guest (AgNPs). The tendency of silver nanoparticles to agglomerate is favored by the architecture of the macromolecular matrices formed by (1-4)-linked β-D-mannuronate and (1-4)-linked α-D-guluronateunits which through the carboxyl groups establish intramolecular hydrogen bonds [45,46] and intermolecular bonds with the hydroxyl and methoxy groups grafted on the aromatic residues of the organic structures in the Ag^0^–ginger nanocomposite.

The composites were analyzed by energy-dispersive X-ray (EDX) method to identify the elemental composition of the materials (Figure 5). EDX spectra of multicomponent gels authenticated the presence of elemental silver (Table 1). All three types of flour have polysaccharides as main components, confirmed by qualitative elemental analysis, recorded by EDX measurements. The difference between them is represented by the content of minerals: Mg in rice, Na, and K in alginate and carob flour, and Ca in all three types of aggregates. Carob flour has the most complex content compared to the other two biopolymers, and in addition, already found in the composition are fibers, tannins, and fatty acids. The presence of sulfur in EDX measurements proves the content of proteins that form aggregates through covalent bonds and disulfide bridges [47].

#### 2.1.4. Thermogravimetric Analysis

By thermogravimetric analysis (Figure 6), we evaluated the stability of multicomponent systems consisting of coated fabrics in comparison with the original cellulosic support. The fabrics present four stages of thermal decomposition, with maximum temperatures directly influenced by the morphological structure of the hydrogel used to finish the fabric. The first stage of decomposition takes place up to 128 °C when the mass loss is about 2–3.4% and is attributed to the processes of water evaporation. The second stage of decomposition takes place up to 232 °C, but in each case, maximum temperature differs depending on the polymer matrix. This stage is attributed to the evaporation processes of the interstitial water sequestered following the hydration processes of the natural polymer matrices. G3 has the lowest mass loss, of about 2%, which correlates with the high content of other bioactive compounds in the composition of the natural matrix and supports the hypothesis of the formation of a hydrogel network with increased density [45]. The organization of the hydrogel resulting from the hydration of the alginate matrix, in a robust network supported by intermolecular hydrogen bonds, makes the maximum temperature of this stage of decomposition the highest, approximately 228 °C, and the mass loss percentage is 2.4%, comparable to G3. The gelation of the starch in G1 leads to a high retention of water in the hydrogel matrix, shown by amass loss of 5.2%, double compared to G2 and G3. The weak starch hydrogel network has the maximum decomposition temperature for this stage at 147 °C, being the lowest among the three samples. The next stage of decomposition takes place around the maximum temperature of 346 °C and is directly related to the decomposition processes of the cellulosic support, where the raw fabric has a mass loss of 85%. For G1 and G2, the maximum temperature is reached at 341 °C, but the mass loss is lower for G1 (76%) compared to 82% in the case of G2. The differences are due to the different content of excipients in the natural polymer matrices confirmed by EDX analysis. The last stage of decomposition is characterized by a greater mass loss for G1 and G3, of 16% and 22%, respectively, in which the auxiliary organic residues of the biopolymer matrix, represented by fatty acids, phospholipids, and polysaccharide residues left from the collapsed hydrogel network are decomposed. The maximum temperature of this stage is the highest in the case of G3 and is influenced by the content of the natural matrix obtained using carob flour. At the end of the thermal decomposition of fabrics covered with hydrogels, a low residue of 0.7–2% was obtained consisting of inorganic compounds resulting from the thermal decomposition of the analyzed multicomponent systems.

#### 2.1.5. Color Parameters in CIEL*a*b* System and Water Contact Angle Measurements

The chromatic characterization of the fabrics finished with natural hydrogels loaded with silver nanoparticles was carried out by comparing the color coordinates L*, a*, b*. The measurements were made in a three-dimensional colorspace consisting of three axes using as parameters 10◦ observer and D65 illuminant. Comparing the coordinates measured for the initial polymer matrix with those of the resulting hydrogel and deposited on the cellulosic support, it was observed that in the case of G1 and G3, coatings with higher brightness, L* = 68.01 and L* = 76.42, respectively, are obtained. A slight change in shade to yellow is obtained for G1 (b* = 4.77), as a result of the hydrogen bonds established between the hydrated matrix and the Ag^0^–ginger nanocomposite. The shade of G3 turns towards blue due to the drastic decrease in b* from 21.45 in the initial matrix to b* = 3.63 for hydrogel. In the case of fabric G2, its brightness L* = 68.81 does not differ much from that of the initial alginate matrix L* = 63.68. The modified hue towards yellow is significant due to the displacement of the value of alginate powder b* = 21.64 to low values b* = 7.04 due to the hydrogen bonds established in the hydrogel matrix, with the AgNP hosting compound and the intermolecular ones established between the hydrogels and the cellulosic fibers [48,49,50].

These intermolecular bonds established between the natural hydrogels and the cotton fabric lead to stable and resistant coatings with relatively good adhesion performance in both dry and wet conditions (Table 2) [51].

The fabrics finished with multicomponent systems based on polysaccharide networks in which AgNPs were embedded are characterized by good hydrophilicity, having contact angles between 75° and 107°, and a high rate of adsorption of water droplets due to the OH groups that were found in the structure of hydrogel matrices.

### 2.2. Spectrophotometric Determination of Silver Ions by the Dithizone Method

Another study was carried out regarding the ability of the gelled matrices deposited on the textile support to release silver ions from the network. The presence of silver ions in the aqueous system was confirmed by Shah’s method [51] based on silver ion complexation with dithizone (DTZ). This is a sensitive method for determining metal ions (Ag^+^, Cd^+^, Hg^+^) by complexing them with DTZ. The concentration of complexed ions is determined spectrophotometrically by measuring the molar absorbance of the solution in the presence of a nonionic or cationic surfactant. The method involves the detection of silver ions complexed with dithizone, a process that can be detected visually due to the change in color from green to brick red. The color change was evaluated by absorption spectroscopy (peaks characteristic of DTZat ~456 and 620 nm) at about 0.97 mM concentration, where the solution obeys Beer’s law (Figure 7). After one hour of hydration of the coated textile samples, the concentration of Ag ions in the aqueous solution was measured spectrophotometrically. Comparing the absorbance recorded for the three samples to the calibration curve for known concentrations of Ag^+^, it was found that the related solution G1 had a concentration of 7.3 ppm [Ag^+^], the G2 solution had 6.8 ppm, while for G3, a concentration of 9.4 ppm [Ag^+^] was found. The results confirmed that Ag^+^ ions can migrate through the gel network and can be released into the external environment, where they can exhibit an antimicrobial effect.

### 2.3. Antimicrobial Study

The antimicrobial activity of silver nanoparticles has been confirmed in a significant number of articles that can be found in the specialized literature [37,38]. In the present work, it was observed how AgNPs keep their antimicrobial properties and their efficiency after being loaded into hydrogel matrices. These aspects are important to know in order to be able to further develop new fields of application or to diversify the delivery matrices of silver nanoparticles.

Specimens from each cotton fabric finished with natural hydrogel loaded with the Ag^0^–ginger nanocomposite were evaluated regarding the antifungal and antibacterial activity by the “Agar diffusion plate test” (Table 3). According to the method presented in the literature [52,53] and in ISO 20645:2004 [54], each fabric sample was placed on a wet culture of microbial cells. The samples were moistened, and the silver ions diffused, reaching the culture of bacteria/fungi, where they affected their growth more or less in relation to the concentration of ions released from the multicomponent system. After 24 h of incubation, the test results showed that specimens G1 and G3 have a moderate antimicrobial activity (at the limit of effectiveness), showing rather a bacteriostatic and fungistatic effect, while for G2, a “strong growth” of microbial cells was observed after the removal of the textile specimen (Figure 8). The results of the antimicrobial activity are in accordance with the measurements obtained when evaluating the capacity to release Ag^+^ from the hydrogel network, where G3 recorded the highest concentration of released ions. The hydrophilic properties of the fabrics favor the adhesion of the biofilm on the surface, and the formation of AgNP agglomerates in the alginate hydrogel matrix determined a weak antimicrobial activity in the case of G2. Considering that the best results were obtained in the case of the carob matrix, the uncharged matrix and the hydrogel system with different loadings of silver nanoparticles were evaluated. Taking into account the literature data regarding the antioxidant and antimicrobial effects given by the content of polyphenols, tannins, and other bioactive compounds [34] of carob flour, it was tested whether the cellulose support covered with carob hydrogel (G3-0) showed antimicrobial properties. Encouraged by the initial results obtained, the increase in the effectiveness of the antimicrobial activity of the fabric was studied if the amount of Ag^0^–ginger nanocomposite were doubled (G3-1). The evaluation after the incubation time was carried out visually and under the microscope, and no antimicrobial activity was observed in the case of G3-0 against either fungi or bacteria (Figure 9a–c). For specimen G3-1, the results showed a lack of growth of microbial cells under the textile substrate. Therefore, G3-1 had a very good antimicrobial activity, best observed against *S. aureus* cells, where a halo formed around the fabric at a distance of 1 mm. However, the enrichment of the carrier matrix with silver nanoparticles, by increasing the amount of Ag^0^–ginger nanocomposite, led to a decrease in the viscosity of the multicomponent system and had the effect of its partial yielding from the fabric, in wet conditions, as can be seen for G3-1 in Figure 8. G3-0 and G3-1 were also evaluated by electron microscopy; the SEM images obtained, shown in Figure 9, confirm the adhesion of microbial cells on the surface of cellulose fibers coated with hydrogelated matrices, while in the case of hydrogel coatings with AgNPs embedded in the network, they are untouched by Gram-positive and Gram-negative cells and fungi.

Considering that textile surfaces evaluated by electron microscopy are very small and to support the results obtained regarding the antimicrobial activity of multicomponent coatings, counts of cells grown on fabrics exposed to Gram-positive bacteria were performed. The comparison of the effectiveness of the antimicrobial activity of G3-type fabrics covered with hydrogel with/without Ag nanoparticles was performed on the most strongly inhibited *S. aureus* strain. For the three specimens (G3-0, G3, G3-1), viable cells were counted on the specimen fabrics after 24 h of contact with the inoculated microorganism. The evaluation of the effectiveness of finished fabrics covered with hydrogel matrices from carob flour with varying content of silver nanoparticles showed that for G3-0 at dilutions of 10^−5^, CFU/mL = 4.32 × 10^8^, and for G3 at dilutions of 10^−5^, CFU/mL = 3.24 × 10^8^, there is a bacteriostatic effect, causing a moderate multiplication of bacteria. In the case of sample G3-1 at the dilution of 2 × 10^−6^, no colonies were observed on the surface of the culture medium, which confirms once again that G3-1 has a bactericidal effect (kills 99.9% of bacteria). Its activity is due to the presence of silver ions that diffused through the network of the hydrogel in which it was loaded (Figure 10).

In this way, it is confirmed that the hydrogel network through its architecture influences the polydisperse system of natural composites, and at the same time, the amount of silver nanoparticles embedded in the gelled matrix affects the antimicrobial properties of the multicomponent system.

## 3. Conclusions

Obtaining multicomponent gels with antimicrobial properties consisted of incorporating Ag^0^–ginger nanocomposites in natural polymer matrices using an eco-friendly method. Ag^0^–ginger nanocomposites result from the phytosynthesis process of reducing silver ions to metallic nanoparticles in the presence of bioactive compounds from ginger extract. By using natural matrices of different plant origin, in which the nanocomposites were loaded at low temperature, three types of multicomponent system were obtained and deposited on the cotton fabric. The finished surfaces showed hydrophilic properties and a good adhesion to the cellulosic substrate. Tests performed on finished textile materials regarding the release of silver ions highlighted the influence of the gelled matrix structure on the migration capacity of the ions from the multicomponent system to the external environment. All fabrics finished with natural composites showed bacteriostatic and fungistatic effects. By optimizing the amount of Ag^0^–ginger nanocomposite loaded in the gelled matrix, fabrics with antimicrobial effect were obtained.

## 4. Materials and Methods

### 4.1. Materials

The extract of active compounds was prepared by mixing 5 g of ginger powder purchased from the local company Organic India (Bucharest, Romania) and 100 mL of alcoholic aqueous solution (30%) under stirring on a magnetic stirrer, at 60 °C for 30 min. The resulting mixture was cooled to 25 °C and separated using a Büchner funnel (Jena, Germany) and Whatman no. 1 filter paper (Merck, Darmstadt, Germany). The obtained yellow liquid was stored in a 100 mL flask and completed with deionized water. In a 100 mL flask, 20 mL of ginger extract from the stock solution and 20 mL of silver nitrate aqueous solution (1 mmol) were added under continuous stirring. The obtained nanoparticles were synthesized according to the method reported by El Refai [38] with a slight modification. The reaction mixture in an equimolar ratio of (1:1) plant extracts to silver nitrate solution was kept under stirring for another four hours at room temperature, when the color of the mixture turned brown. Thus, silver nanoparticles in ginger extract (Ag^0^—ginger) were synthesized through an eco-friendly method.

Natural polymer gels were obtained by dissolving 2 g of alginate/6 g of rice flour/6 g of carob flour bought from a local supermarket in 100 mL of deionized water, each under stirring and at a temperature of 80–90 °C, for one hour. Over 8 g of the gelled matrices were added to 2 g of Ag^0^–ginger nanocomposite, stirring continuously. After incorporating the solution containing silver particles into the gelled matrices, three types of homogeneous gels were obtained, and they were deposited on 2 g cotton fabric by the screen printing method. The finished fabrics were dried at 25 °C for 2 h, the nat 105 °C for 1 h in a thermofixation oven (Biobase, Shandong, China). The fabric used as a support for the natural hydrogels was100% cotton (Matasea Romana, Romania) with the following characteristics: specific weight = 161.98 g/m^2^, 79.4 ends, 65.4 picks, yarn count of 23.4 × 22 tex; it was originally scoured and chemically bleached.

### 4.2. Methods

The cellulosic fabrics finished with the natural composites loaded with silver nanoparticles were analyzed to confirm the presence of silver by recording XRD diffractograms in the range 2θ, 2–90° with a RigakuSmartLab equipment (Rigaku Corporation, Tokyo, Japan). The diffractograms were interpreted using the PDXL software (ver. 2.7.2.0, Rigaku Corporation, Tokyo, Japan), provided by Rigaku, and the identification of the silver and silver oxide present at the textile surface was performed by comparison with the ICDD (International Centre for Diffraction Data) database entries. The structural analysis of the three types of biopolymers and the interactions between their hydrogel matrix and cotton was carried out by FTIR measurements. Spectra were recorded in the range 400 to 4000 cm^−1^, making 128 accumulations at a resolution of 4 cm^−1^ with a JASCO FT-IR 6300 instrument (Jasco Int. Co. Ltd., Tokyo, Japan), which was equipped with a Specac ATR Golden Gate (Specac Ltd., Orpington, UK) with KRS5 lens. SEM images of coated fabrics with natural hydrogels and those inoculated in microbiologic medium were obtained with a TM4000Plus equipment (HITACHI, Tokyo, Japan) using the following operating conditions: acceleration voltage of 10 kV and magnifications up to 1800×. The thermal stability of the natural composites was studied by thermogravimetric analysis (TGA) performed with a TGA Q5000IR instrument (TA Instruments, New Castle, DE, USA). The quantity of the samples subjected to thermal decomposition was about 3.5–4 mg each, and it was weighed in platinum pans and analyzed under the following conditions: heating ramp 10 °C/min up to 600 °C, Synthetic Air (99.999%) used as purge gas at a 50 mL/min flow rate. To evaluate the colors of finished textile samples and raw flours, total color differences were measured in the CIELAB system, using a 10° standard observer and illuminant D65, with V570 UV-VIS-NIR equipment (Jasco Int. Co. Ltd., Tokyo, Japan) equipped with an integrating sphere (JASCO ILN-472 (150 mm)), using spectralon as reference material. The hydrophilic properties of the coatings were evaluated by measuring the water contact angle with a CAM 200 (KSV Instruments, Helsinki, Finland) equipped with a high-resolution camera (Basler A602f, Basler, Ahrensburg, Germany) and an autodispenser. Water contact angles were measured in air, at 25° C and ambient humidity, 2 s after the drop contacted the surface of the coatings. Drops of 6 µL deionized water were dispensed on each sample, and the value of the reported water contact angles was the average of six measurements. Natural coated fabrics were evaluated for rubbing fastness according to ISO-105X12 [51].

The evaluation of the antibacterial activity of cotton finished with natural composites was carried out on bacterial strains, *Staphylococcus aureus* (ATCC 25923) and *Escherichia coli* (ATCC 25922), and strain of fungi, *Candida albicans* (ATCC 10231), from the Microbial Collection of National Research and Development Institute for Chemistry and Petrochemistry-ICECHIM. For the evaluation of the antimicrobial activity, we used the “Agar diffusion plate test” method with coated fabric samples (2 × 2 cm^2^) on the Sabouraud medium for fungi and Muller Hinton medium for bacteria, respectively, inoculated into the cloth with the tested microorganisms. Antimicrobial activity was evaluated according to ISO 20645:2004 [54] by measuring the diameter of the clear area (halo) that appeared around the textile specimens. The working inoculum was represented by a suspension made from a fresh culture in AFS (sterile physiological water), developed on a solid TSA (Tryptic Soy Agar) medium, with a density of 1.5 × 10^8^ CFU/mL, nephelometrically adjusted (McFarland standard 0.5 = 1.5 × 10^8^ CFU/mL). The plates were incubated for 24 h at the optimal temperature (28 °C for fungi and 37 °C for bacteria) and then the cell cultures were examined. Following the evaluation of cellular microorganisms, it was observed that the best results were in the case of *S. aureus* cultures and the tests were continued by counting viable cells. The viable cells on the textile samples were counted after 24 h of contact of the textile sample with the inoculated microorganism by the method presented below. The textile sample was transferred into 20 mL of AFS and vortexed to help detach the bacterial cells from the textile support. The supernatant was serially diluted up to 10^−5^ dilution and inoculated with 100 µL of each on TSA medium. After 24 h of incubation, the colonies were counted. The number of viable bacteria was reported as colony forming units, CFU/mL.

## Data Availability

The data are not publicly available due to their containing information that could compromise the privacy of research participants.

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
