# Peer review of "Ag^0^–Ginger Nanocomposites Integrated into Natural Hydrogelated Matrices Used as Antimicrobial Delivery Systems Deposited on Cellulose Fabrics"

_gels, 2024, doi:10.3390/gels10020106_

Round 1
Reviewer 1 Report
Comments and Suggestions for Authors
Author Response
Dear reviewer,
Thank you for your advice and requirements which will improve the quality of our manuscript.
We have completed the document according to the suggestions you gave us, as follows:
Pg1, al.21, “antifungal coatings” it was replaced with “antimicrobial coatings”.
Pg.3, al.86, “…matrices based on rice flour, carob flour and alginate (scheme 1, a,b,c)…”
It was changed it in acording with the scheme 1, namely rice flour, alginate and carob flour (scheme 1, a,b,c).
Pg.9.Fig.7. The content of fig. 7 has been improved by adding new pictures and additional explanations.
Pg.8, al. 254 "Spectrophotometric determination of silver ions by the dithizone method" was delimited by the subchapter "Antimicrobial activity" and is discussed outside of it.
Pg.9. There was a name confusion between the viable cell counting method and the CMI method. The error has been removed.
Pg.9.al.283-285
Fig.7 has been modified, in that the image of the textile specimen and the image after its removal from the microorganism cultures have been placed. In this way, the bacteriostatic or fungistatic effect of the samples can be highlighted, the decrease or absence of the microorganism population being observed.
Pg.9. al.288-291
In order to present the antimicrobial activity of the tested fabrics more clearly, a table was inserted with the results obtained following the visual evaluation of the microorganism cultures after the removal of the textile specimens.
Table 3. Evaluation of microbial cells grown under the textile specimens
|
Sample |
Growth of bacteria/fungi under the specimen |
||
|
|
S. aureus |
E. coli |
C. albicans |
|
G1 |
none/slight |
none/slight |
slight |
|
G2 |
heavy |
moderate/heavy |
moderate/heavy |
|
G3 |
none |
none/slight |
none |
Adopted from ISO 20645:2004: none=good effect; slight=limit of efficacy; moderate=insufficient effect; heavy=insufficient effect.
Pg.9. al. 291-294 Mention the results that demonstrate the antimicrobial activity of the G3-0 matrix.
The text has been modified and completed ”Taking into account the literature data regarding the antioxidant and antimicrobial effects given by the content of polyphenols, tannins and other bioactive compounds [34] of carob flour, it was tested if the cellulose support covered with carob hydrogel (G3-0) shows antimicrobial properties.
Pg.9. al. 291-294 The assessment of "weak" antimicrobial activity must be given a value. How did the authors assess visually? The diameter of the inhibition zone is reported in the methods, but these results are not presented in the manuscript.
The manuscript has been completed:
“Specimens from each cotton fabric finished with natural hydrogel loaded with the Ag0-ginger nanocomposite were evaluated regarding the antifungic and antibacterial ac-tivity by “Agar diffusion plate test”(table 3). According to the method presented in the litherature [52, 53] and in ISO 20645;2004, each fabric sample was placed on wet culture of microbial cells.”
Pg.9. al. 299-300
The text has been modified and completed:
“G3-1 had a very good antimicrobial activity, best observed against S. aureus cells, where a halo was formed at a distance of 1 mm from the edge of the fabric.”
Pg.9. al. 301-303; Pg.10.al. 304-305
The text has been modified and completed:
“G3-0 and G3-1 were also evaluated by electron microscopy, the SEM images obtained in figure 9, confirm the adhesion of microbial cells on the surface of cellulose fibers covered with hydrogelatedmatrices, while the hydrogel coatings with AgNps embedded in the network they are untouched by Gram-positive, Gram-negative cells and fungi.”
Pg.10.al. 309. Pg.10.al.310-312 Pg.10.al.313-319
As I mentioned, there was an error in expression regarding the method of counting viable cells (not the CMI method), and the text was completed as follows:
“Considering that textile surfaces evaluated by electron microscopy are very small and to support the results obtained regarding the antimicrobial activity of multicomponent coatings, counts of cells grown on fabrics exposed to Gram-positive bacteria were performed.The comparison of the effectiveness of the antimicrobial activity of G3 type fabrics covered with hydrogel with/without Ag nanoparticles was performed on the most strongly inhibited S. aureus strain. For the three specimens (G3-0, G3, G3-1), viable cells were counted on the specimen fabrics after 24 hours of contact with the inoculated microorganism. The evaluation of the effectiveness of finished fabrics covered with hydrogel matrices from carob flour with varying content of silver nanoparticles, showed that for G3-0 at dilutions of 10‒5, CFU/mL = 4.32 × 108, and for G3 at dilutions of 10‒5, CFU/mL = 3.24 × 108 have a bacteriostatic effect, causing a moderate multiplication of bacteria. In the case of sample G3-1at the dilution of 2x10-6, no colonies were observed on the surface of the culture medium, which confirms once again that G3-1 has a bactericidal, effect (kills 99.9% of bacteria ), its activity due to the presence of silver ions that diffuse through the network of the hydrogel in which it was loaded (figure 10).
In this way, it is confirmed that the hydrogel network through its architecture influences the polydisperse system of natural composites and at the same time the amount of silver nanoparticles embedded in the gelled matrix affects the antimicrobial properties of the multicomponent system.”
Pg.10.fig.9
The figure has been completed.
Figure 10. Evaluation by the method ofcounting cells grown after exposure to Gram-positive bacteriaof cotton finishedwith carob hydrogel matrices (G3-0) and loaded with different concentrations of Ag0-ginger nanocomposite (G3, dilutions of 10‒5 and G3-1, dilutions of 2x10‒6).
Due to the recommendations of the other reviewers, the text was slightly modified.
Best regards,
Valentin Raditoiu

Reviewer 2 Report
Comments and Suggestions for Authors
Dear Authors,
The manuscript presents an experimental study on Ag0 -ginger nanocomposites integrated into natural hydrogelated matrices used as antimicrobial delivery systems deposited on cellulose fabrics. For that purpose, Fourier Transform Infrared Spectroscopy (FTIR), X-ray Diffraction (XRD), Scanning Electron Microscopy and Energy-Dispersive X-ray (SEM-EDX) analysis, thermogravimetric measurements and antimicrobial studies were performed. The work is very interesting and very well written. The method used in the research is clear and understandable. Undoubtedly, the authors master their approach well and obtained interesting results. However, below are questions and remarks that should be corrected before the manuscript could be consider for publication in Gels.
1. Please, explain in 2-3 sentences what the Shah's method is (line 260). It would be more comfortable for the reader than looking for this in literature.
2. Please, explain in the caption below Figure 6 what does it mean the colour lines in a picture (a) and what is in (b) picture.
1. Lines 91-92: “much” is repeated
Line 126: FTIR with capital letters
Author Response
Dear reviewer,
Thank you for your advice and requirements which will improve the quality of our manuscript.
We have completed the document according to the suggestions you gave us, as follows:
- Please, explain in 2-3 sentences what the Shah's method is (line 260). It would be more comfortable for the reader than looking for this in literature.
The manuscript has been completed:
“This is a sensitive method for determining metal ions (Ag+, Cd+, Hg+) by complexing them with DTZ. The concentration of complexed ions is determined spectophotometrically by measuring the molar absorbance of the solution in the presence of a nonionic or cationic surfactant. In the present study.”
- Please, explain in the caption below Figure 6 what does it mean the colour lines in a picture (a) and what is in (b) picture.
The manuscript has been completed:
“Figure 7. Absorption spectra of dithizone solution (DTZ) and effect of silver-dithizone complex on the absorbance at lmax 620 nm(a). Calibaration graph of DTZ standard solution (0.97 mM) with increasing amounts of Ag+ (0−40 ppm) (b).“
- Lines 91-92: “much” is repeated
The text has been corrected.
- Line 126: FTIR with capital letters
The text has been corrected.
Due to the recommendations of the other reviewers, the text was slightly modified.
Best regards,
Valentin Raditoiu

Reviewer 3 Report
Comments and Suggestions for Authors
The manuscript titled "Ag0-ginger nanocomposites integrated into natural hydrogelated matrices used as antimicrobial delivery systems deposited on cellulose fabrics" explores the development of antimicrobial coatings for textile materials. The study focuses on environmentally friendly methods for creating coatings with antimicrobial properties, specifically utilizing natural polymers loaded with silver nanoparticles.
The article has left a positive impression, indicating that the authors have conducted extensive experimental work. They have described the results, drawing on contemporary literature.
The article is recommended for publication after addressing a few minor comments.
In general, the language needs moderate improvement (see, for instance lines 82-85, 95-97 and further), check formatting, e.g. in lines 268-270. Ag+ superscript is missed, and correct the typos, for instance, in Figure 7: Fugstatic, Atifungic, line 126: FTIR.
Line 21. Are the coatings only antifungal, or are they also antimicrobial?
Scheme 1. At what stage does gelation occur? Is the gel already formed in the initial flask where ginger is added? Following the logic of the scheme, was the system dehydrated after the nanoparticles were obtained?
Lines 165-167. Please improve the clarity of the statement. It is written that Ag NPs were uniformly dispersed throughout the matrix. THUS, the type of polysaccharides and some treatments influenced rheology, viscosity, gelation process, and water adsorption. Please explain how these properties are connected with the distribution of NPs.
Lines 172-173. Paraphrase this part: "strengthened by bonds of hydrogen with embedded silver nanoparticles." The term "bonds of hydrogen" sounds incorrect here, especially with embedded NPs.
Line 177. Here and in several other parts of the manuscript, the materials G1-3 are called biodegradable. Was it proved somehow? I am confident that the fabrics, as well as the hydrogels, are truly biodegradable. But is the composite with Ag also biodegradable? Also, how is this property connected to EDX?
Line 181 and in other places. Where and how was the supramolecular nature of the aggregates confirmed?
Lines 233-234, 239-241. Can the authors provide some literature to support the statement about the connection between changes in hydrogen bonds and the color of the finished fabric?
Line 248. Have you checked the wash resistance of the coated fabrics?
Line 349. Please correct the description of the protocol: "Take 20 mL of ginger extract <...> and add 20 mL of..." Lines 358 and 359, the same note.
Line 356 and further. Is it a gel (as it is written in the article above) or is it a paste?
Lines 363-365. What color did the initial cotton fabric have?
Lines 397-399 and in the text - the names of the species in italics.
Lines 405. The agar media is solid in the given temperature range. The fabric was also dry, as far as I can see from the description. Did the authors moisturize the media somehow to provoke the Ag+ ions?
Comments on the Quality of English LanguageModerate editing of English language required
Author Response
Dear reviewer,
Thank you for your advice and requirements which will improve the quality of our manuscript.
We have completed the document according to the suggestions you gave us, as follows:
Line 21. Are the coatings only antifungal, or are they also antimicrobial?
The text has been corrected.
Scheme 1. At what stage does gelation occur? Is the gel already formed in the initial flask where ginger is added? Following the logic of the scheme, was the system dehydrated after the nanoparticles were obtained?
The Scheme 1 has been modified and completed:
Lines 165-167. Please improve the clarity of the statement. It is written that Ag NPs were uniformly dispersed throughout the matrix. THUS, the type of polysaccharides and some treatments influenced rheology, viscosity, gelation process, and water adsorption. Please explain how these properties are connected with the distribution of NPs.
The text has been modified and completed:
“For the study of the natural matrices of the delivery systems, three types of flour characterized by different saccharide structural sequences were chosen (figure 3). These, following the gelation process, generate networks with their own architecture and specific characteristics for the resulting gels. Thus, starch, the major component in rice, has the property of swelling and can easily retain interstitial water [28], while the carboxylic groups contained on the polysaccharide chain favor cross-linking processes in the presence of divalent ions[30].Carob flour has a larger ramified polysaccharide sequence, called galactomannan, which, together with the proteins and fibers it contains, manifests itself as gum[26]….. The architecture of the gels resulting from the hydration of rice and carob polysaccharides favors a good dispersion of the nanocomposite loaded in the natural polymer network.”
Figure 3. Polysaccharide sequences that constitution the natural polymer matrices from rice flour (a), sodium alginate (b) and carob (c).
Lines 172-173. Paraphrase this part: "strengthened by bonds of hydrogen with embedded silver nanoparticles." The term "bonds of hydrogen" sounds incorrect here, especially with embedded NPs.
The text has been modified and completed:
“The tendency of silver nanoparticles to agglomerate is favored by the architecture of the macromolecular matrices formed by (1-4)-linked b-D-mannuronate and (1-4)-linked a-D-guluronateunits which through the carboxyl groups contained establish intramolecular hydrogen bonds[45,46] and intermolecular bonds with the hydroxyl and methoxy groups grafted on the aromatic residues of the organic structures in the Ag0-ginger nanocomposite.”
Line 177. Here and in several other parts of the manuscript, the materials G1-3 are called biodegradable. Was it proved somehow? I am confident that the fabrics, as well as the hydrogels, are truly biodegradable. But is the composite with Ag also biodegradable? Also, how is this property connected to EDX?
It was not expressly proven that the materials are biodegradable, therefore the text was corrected.
Line 181 and in other places. Where and how was the supramolecular nature of the aggregates confirmed?
The supramolecular nature of the aggregates was not proven, therefore the text was corrected.
Lines 233-234, 239-241. Can the authors provide some literature to support the statement about the connection between changes in hydrogen bonds and the color of the finished fabric?
The specialized literary references have been added.
- Glowacki, E. D.; Irimia-Vladu, M.; Bauerb, S.; Sariciftci, N. S. Hydrogen-bonds in molecular solids – from biological systems to organic electronics. J. Mater. Chem. B,2013, 1, 3742. DOI: 10.1039/c3tb20193g.
- Raduly, F.M.; Rădițoiu, V.; Rădițoiu, A.; Frone, A.N.; Nicolae, C.A.; Răut, I.; Constantin, M.; Grapin, M. Multifunctional Finishing of Cotton Fabric with Curcumin Derivatives Coatings Obtained by Sol–Gel Method. Gels2023, 9, 369. https://doi.org/10.3390/gels9050369.
- Fleischmann, C.; Lievenbrück, M.; Ritter, H. Polymers and Dyes: Developments and Applications. Polymers2015, 7, 717-746. https://doi.org/10.3390/polym7040717
Line 248. Have you checked the wash resistance of the coated fabrics?
The authors did not perform washing resistance tests of the finished textiles. These properties will be evaluated in future studies aimed at specific applications for making clothes.
Line 349. Please correct the description of the protocol: "Take 20 mL of ginger extract <...> and add 20 mL of..." Lines 358 and 359, the same note.
The text has been modified and completed:
“The extract of active compounds was prepared by using 5g of ginger powder purchased from the local company Organic India and 100 mL of alcoholic solution (30%) stirring on a magnetic stirrer, at 60 °C for 30 min The resulting mixture was cooled to 25 0C and separated by a Büchner funnel with Whatman no. 1 filter paper. The obtained yellow liquid was stored in a 100 mL flask and completed with deionized water.In a 100 mL flask,20 mL of ginger extract from the stock solution were placed, 20 mL of aqueous silver nitrate solution (1 mmol) were added, under continuous stirring.”
Line 356 and further. Is it a gel (as it is written in the article above) or is it a paste?
The text has been corrected.
Lines 363-365. What color did the initial cotton fabric have?
The cotton fabric was initially white, characterized by the color coordinates: L* = 82.94; a* = -0.07;
b* = -1.14 (table 2). Chemical bleaching is usually a pretreatment to which the fabric is subjected before the dyeing process.
Lines 397-399 and in the text - the names of the species in italics.
The text has been corrected.
Lines 405. The agar media is solid in the given temperature range. The fabric was also dry, as far as I can see from the description. Did the authors moisturize the media somehow to provoke the Ag+ ions?
The text has been completed:
“According to the method presented in the literature [52, 53] and in ISO 20645;2004, each fabric sample was placed on wet culture of microbial cells. The samples are moistened and the silver ions diffuse reaching the culture of bacteria/fungi where they affect their growth more or less in relation to the concentration of ions released from the multicomponent system.”
Due to the recommendations of the other reviewers, the text was slightly modified.
Best regards,
Valentin Raditoiu
